# Characterization of Giant Magnetostrictive Materials Using Three Complex Material Parameters by Particle Swarm Optimization

**DOI:** 10.3390/mi12111416

**Published:** 2021-11-18

**Authors:** Yukai Chen, Xin Yang, Mingzhi Yang, Yanfei Wei, Haobin Zheng

**Affiliations:** College of Electrical and Information Engineering, Hunan University, Changsha 410082, China; chenyukai@hnu.edu.cn (Y.C.); mingzhiyang@hnu.edu.cn (M.Y.); wyf_8899@163.com (Y.W.); zhenghaobin@hnu.edu.cn (H.Z.)

**Keywords:** giant magnetostrictive material, complex parameters, losses, transducer, lumped parameter model, particle swarm optimization (PSO) algorithm

## Abstract

Complex material parameters that can represent the losses of giant magnetostrictive materials (GMMs) are the key parameters for high-power transducer design and performance analysis. Since the GMMs work under pre-stress conditions and their performance is highly sensitive to pre-stress, the complex parameters of a GMM are preferably characterized in a specific pre-stress condition. In this study, an optimized characterization method for GMMs is proposed using three complex material parameters. Firstly, a lumped parameter model is improved for a longitudinal transducer by incorporating three material losses. Then, the structural damping and contact damping are experimentally measured and applied to confine the parametric variance ranges. Using the improved lumped parameter model, the real parts of the three key material parameters are characterized by fitting the experimental impedance data while the imaginary parts are separately extracted by the phase data. The global sensitivity analysis that accounts for the interaction effects of the multiple parameter variances shows that the proposed method outperforms the classical method as the sensitivities of all the six key parameters to both impedance and phase fitness functions are all high, which implies that the extracted material complex parameters are credible. In addition, the stability and credibility of the proposed parameter characterization is further corroborated by the results of ten random characterizations.

## 1. Introduction

Giant magnetostrictive materials (GMM) such as Terfenol-D are important smart materials for underwater acoustic transducers [1,2]. When modeling and designing high-power underwater transducers, knowledge of the complex parameters of characteristic materials is needed to predict the performance and iteratively optimize the design [3,4]. One of the challenges faced by transducer designers is a lack of accurate and reliable characteristic data regarding the properties of GMMs [5]. Particularly, energy losses in smart materials remarkably affect the important characteristics of a high-power transducer, such as the electrical impedance or the amount of heat generated. Accurately characterizing the losses of smart materials is very demanding [6,7,8].

The parameter characterization methods of magnetostrictive materials can be mainly divided into two categories, namely, direct measurement methods and impedance analysis methods [9]. A direct measurement method refers to the method of extracting material parameters from measured hysteresis loops and magnetostrictive curves [10]. This method suits the evaluation of material quality, and the measurement involves only the local position of the material specimen; however, achieving sufficient accuracy for the measured parameters requires instruments with high accuracy, which is normally expensive and introduces measurement noise.

In the field of piezoelectric materials, impedance analysis methods have become the most effective methods to characterize the complex parameters of equivalent characteristic materials [11]. Using complex material parameters, the losses can also be characterized. This is normally achieved by intelligent algorithms to minimize the difference between the harmonic response data measured by the impedance analyzer and the simulated data in order to characterize the material parameters. The IEEE Standard on Piezoelectricity uses a lumped parameter model to describe the impedance properties in a one-dimensional mode and calculates the real parameters of the material by measuring the impedance curve, but this method cannot directly describe the material losses because it does not use complex material parameters [12]. In fact, Sherrit et al. have proved that a lumped parameter impedance model with complex material parameters is effective, efficient, and can fit impedance data with high accuracy and be used to calculate the complex parameters of materials [13]. Wild et al. [14,15] developed a 1D equivalent circuit or 3D FEM and the impedance curve measured to characterize the complex parameters under the radial vibration mode of the piezoelectric material. Sun et al. [16] successfully extracted the parameters of the high-loss piezoelectric composite material. Moreover, these studies show that the extraction of the imaginary parts (losses) of the complex parameters are more challenging than the real parts. Jonsson et al. [17] extracted the full parameter matrix of the material by a finite element model; however, this powerful characterization method is time-consuming.

The characterization of GMMs is more challenging compared to that of piezoelectric materials. One of the key issues is that the performance of GMMs is very sensitive to prestress and magnetic bias [10]. A recent study of electrical bias and pre-stress effects on the loss factors has provided a better understanding of the microscopic loss mechanism in piezoelectric materials and can facilitate a better finite element analysis on device designing [18]. This is also true for GMMs. It is necessary to introduce a mechanical structure to apply pre-stress to the material and extract material complex parameters under different pre-stress conditions. In addition, GMMs have an eddy current effect that varies with frequency, so they have a more complicated loss mechanism than piezoelectric materials. Dapino et al. [19] adopted the theory of an electroacoustics model based on small-signal excitation and analyzed the dynamic magneto-mechanical characteristic parameters of Terfenol-D under different working conditions by measuring the impedance curve and output displacement of a longitudinal vibrating transducer. Luke et al. [20] refer to the method proposed by Dapino to characterize Galfenol under specific working conditions; however, this method relies on the measured output displacement. In addition, this ignores the losses. Greenough et al. [21,22] established a plane wave model of a longitudinal GMM transducer using complex parameters to represent losses in the material, and extracting key parameters by use of a simulated annealing (SA) algorithm to identify the experimental impedance measurement results under the free-stand state. After that, Greenough [23] further extracted material parameters under different prestress by the same method; however, the influence of the mechanical structure on the parameter characterization is not mentioned. The extracted imaginary parts of complex parameters sometimes turned to positive values under small signal excitations, implying an abnormal dissipation factors tangent [24].

A particle swarm optimization (PSO) algorithm is an efficient parameter identification algorithm, and its effect has been verified in the parameter characterization of electric impedance model [16,25]. Sun et al. [16] used PSO, SA, and Gauss–Newton algorithms to characterize the complex parameters of piezoelectric materials with the thickness vibration mode and showed that the Gauss–Newton algorithm relies heavily on the selection of initial values. Further, the identification results have relatively large and unstable relative errors. The SA algorithm is more dependent on the annealing parameters and annealing time strategy. In contrast, the PSO algorithm can search in a larger parameter range, and quickly converge to the optimal solution when compared to the SA and Gauss–Newton algorithms, so it is more reliable for the characterization of unknown material parameters.

Consequently, an optimized complex parameters extraction routine for GMMs is proposed in this paper while considering three material losses, i.e., hysteresis losses, elastic losses, and piezomagnetic coupling losses under the longitude vibration mode. The purpose of this study is to investigate a method that can stably characterize the complex parameters of GMMs under different pre-stress conditions. The key improvement is to measure and calculate the structural damping and contact damping of the parameter characterization device and apply the data to confine the parametric variance range of material losses. The proposed method is based on a lumped parameter model containing the three losses and uses a PSO algorithm to minimize the root mean square error (RMSE) between the experimental impedance data and simulation data to extract the real parts of the material parameters, and then by minimizing the RMSE between the experimental phase data and the simulation data to extract the imaginary parts. The global sensitivity analysis demonstrates the importance of using the phase data and measuring structural damping and contact damping for parameter characterization. Comparing with the traditional method, the sensitivity of the three losses has been greatly improved. Finally, the complex parameters were randomly characterized ten times, which further confirmed the stability of the method.

## 2. Characterization Methodology

### 2.1. Complex Parameters of GMM

According to the loss mechanism of a GMM [26], there are three types of losses under actual working conditions, namely, hysteresis losses, elastic losses, and piezomagnetic coupling losses. Similar to piezoelectric materials, the small losses (dissipation factor tangent ≪0.1) of a GMM can also be regarded as disturbances and may be introduced into phenomenological equations as “complex physical constants”, which is mathematically equivalent to the role of “dissipation functions” [27]. Consequently, we introduced the complex parameters of relative permeability $\mu_{33}^{\sigma *}$, elastic compliance $S_{33}^{H*}$, and piezomagnetic constant $d_{33}^{*}$ into the linear piezomagnetic constitutive equation of GMM to yield delay-time-related small losses. The complex parameters of longitudinal vibration mode are shown in Equation (1).
(1)$$\left\{ \begin{array}{l} \mu_{33}^{\sigma *}=\mu_{33}^{\prime }+j\mu_{33}^{\prime \prime }\\ S_{33}^{H*}=S_{33}^{\prime }+jS_{33}^{\prime \prime }\\ d_{33}^{*}=d_{33}^{\prime }+jd_{33}^{\prime \prime } \end{array}\right.$$

### 2.2. The Longitudinal Transducer

The structural diagram of a longitudinal GMM transducer is shown in Figure 1. During the operation, the transducer is installed on a shock absorber table to simulate an infinite base mass and the base was actually a seismic mass. A GMM rod with a diameter of 20 mm and a length of 100 mm was used as the driving material. To reduce the eddy current, the rod was cut to 9 slots and the slots were filled with liquid epoxy resin. A photo of the rod is shown in Figure 2.

A 940-turn AC solenoid provided an excitation magnetic field for the rod. A 1540-turn DC solenoid was used to provide a DC bias magnetic field for the rod. Generally speaking, the bias magnetic field provided by the DC solenoid is more uniform along the rod axis than the bias magnetic field provided by the permanent magnet, which is crucial for the material parameter characterization. The closed magnetic circuit of the transducer consisted of a cylindrical magnetic column at both ends of the rod, an outer cylinder, a housing base and an upper end cover. A closed magnetic circuit can minimize the leakage flux and make the magnetic flux in the bar more uniform. The magnetic columns, housing base, and the upper cover were made of soft iron after a slitting treatment, and the outer cylinder was wound by a 0.35 mm silicon steel sheet whose eddy losses are trivial. Spring washers were used to pre-stress the rod. A ring pressure sensor was used to measure the prestress of the transducer.

### 2.3. The Lumped Parameter Model for the Transducer

The lumped parameter model for the transducer is shown in Figure 3. *E* represents the input voltage of the transducer, *I* represents the input current, *Z*_e_ is the blocked electrical impedance, *Z*_t_ is the mechanical impedance, *V* is the output speed, *F* is the output force on the displacement plunger, and *T*_em_ and *T*_me_ stand for the transduction terms “electrical due to mechanical” and “mechanical due to electrical”, respectively. The variables are all variables in the frequency domain. The related linear conversion equation has the following form:(2)$$E=Z_{e}I+T_{\mathrm{em}}V$$
(3)$$F=T_{\mathrm{me}}I+Z_{t}V$$

The transducer’s electrical impedance frequency response function *Z* is given as follows:(4)$$Z=\frac{E}{I}=Z_{e}-\frac{T_{\mathrm{em}}T_{\mathrm{me}}}{Z_{t}}$$

A GMM under an alternating magnetic field would generate eddy current losses. According to [28], the cut-off frequency *f**_c_* of the GMM rod is 30 kHz, which is much greater than the working frequency *f*. In this case, the eddy current factors can be described as per [29]:(5)$$\left\{ \begin{array}{l} \chi_{r}=1-\frac{1}{48}{\left(\frac{f}{f_{c}}\right)}^{2}+\frac{19}{30720}{\left(\frac{f}{f_{c}}\right)}^{4}+\ldots \\ \chi_{i}=\frac{1}{8}\left(\frac{f}{f_{c}}\right)-\frac{11}{3072}{\left(\frac{f}{f_{c}}\right)}^{3}+\frac{473}{4343680}{\left(\frac{f}{f_{c}}\right)}^{5}+\ldots \end{array}\right.$$

The equivalent permeability, which includes the eddy current losses, can be expressed as follows:(6)$$\mu_{33}^{\sigma *}=\mu_{33}^{\prime }(\chi_{r}+j\chi_{i})+j\mu_{33}^{\prime \prime }$$

The $k_{33}^{*}$ magneto-mechanical coupling is defined as follows:(7)$$k_{33}^{*}=\sqrt{{\left(d_{33}^{2}\right)}^{*}/\mu_{33}^{\sigma *}S_{33}^{H*}}$$

In Figure 3, the blocked electrical impedance *Z*_e_ is expressed as follows:(8)$$Z_{e}=R_{0}+j\omega L_{G}^{*}$$
where $L_{G}^{*}=\;(R_{g1}+j\omega L_{g})/j\omega$ represents the equivalent inductance include hysteresis and eddy current losses of electrical part, $R_{g1}=-\omega (\chi_{i}+\mu_{33}^{\prime \prime }/\mu_{33}^{\prime })L_{b}$ and $L_{g}=\chi_{r}L_{b}$.

$L_{b}=(1-{\left(k_{33}^{*}\right)}^{2})\mu_{33}^{\prime }N^{2}A/l$ represents an approximation of the inductance of a wound wire solenoid when the transducer is in a blocked state. *N* and *R*_0_ represent the number of turns and the DC impedance of the AC excitation solenoid, respectively. *A* and *l* represent the cross-section and the length of the rod, respectively.

The mechanical impedance *Z*_t_ is expressed as follows:(9)$$Z_{t}=j\omega M_{t}+(K_{\mathrm{spr}}+K_{G}^{*})/j\omega +R_{d}+R_{f}$$
where *M*_t_ refers to the equivalent mass of transducer, *K*_spr_ represent the equivalent stiffnesses of the pre-stress spring washers, $K_{G}^{*}=j\omega (R_{g2}+1/j\omega K_{g})$ represents the stiffness of the GMM, include elastic losses $R_{g2}=A/(\omega lS_{33}^{\prime \prime })$ and $K_{g}=A/lS_{33}^{\prime }$, and *R*_d_ and *R*_f_ refer to the damping of the displacement plunger and the contact damping of the contact surfaces, respectively.

The transduction coefficient between the electrical and mechanical parts of the transducer is given as follows:(10)$$T_{\mathrm{em}}=-T_{\mathrm{me}}=\frac{NA(d_{33}^{\prime }+jd_{33}^{\prime \prime })}{(S_{33}^{\prime }+jS_{33}^{\prime \prime })l}$$

In summary, the electrical impedance equation of transducer *Z* can be obtained based on the improved parameter model by putting Equations (8)–(10) into Equation (4):(11)$$Z=R_{0}+j\omega \left\{\left[\mu_{33}^{\sigma *}\left(1+{\left(k_{33}^{*}\right)}^{2}(\frac{K_{G}^{*}}{j\omega Z_{m}}-1)\right)\right]\frac{N^{2}A}{l}\right\}$$

The equation of phase *P* can also be expressed as follows:(12)$$P=\arctan(\frac{\mathrm{img}(Z)}{\mathrm{real}(Z)})\frac{180}{\pi }$$
where real(Z) and img(Z) represent the real and imaginary parts of electrical impedance Z, respectively. To estimate the effective magneto-mechanical coupling coefficient $k_{\mathrm{eff}}^{*}$ of the rods, which is affected by mechanical stiffness and flux linkage, can be calculated as per [30]:(13)$${\left(k_{\mathrm{eff}}^{2}\right)}^{*}=\frac{k_{M}^{2}{\left(k_{33}^{2}\right)}^{*}K_{G}^{*}}{K_{G}^{*}+K_{\mathrm{mps}}(1-{\left(k_{33}^{2}\right)}^{*})}$$

In Equation (12), $k_{M}^{}$ represents the leakage flux of the transducer and setting its value as $k_{m}^{2}$ = 0.9205 as per [31].

### 2.4. The Proposed Optimization Method

The essence of parameter identification is the process of using an intelligent optimization algorithm to find the minimum value of an objective function. In this paper, PSO is used to iterate repeatedly to find the complex material parameters in the improved lump parameter model to fit the experimental curves.

The operative process of the PSO is shown in Figure 4. Firstly, the positions and velocities of the particles are initialized, i.e., give each particle a random initial position and velocity and then calculate the fitness function value of each particle according to the fitness function Equation (14) or (15). Then, the fitness function value corresponding to the current position of each particle and the historical best position is compared, the individual optimal value of the particle is updated, and the global optimal value is obtained. Subsequently, the positions and velocities of the particles are updated, and then the fitness function value is calculated. Finally, it is judged whether the termination condition is satisfied, and if the termination condition is satisfied, the identification result is outputted. If the conditions are not met, then the process continues to iterate.

In this research, three different methods of material parameter extraction based on PSO were investigated. A flow chart of all three methods is shown in Figure 5. Method 1 represents the classic method [23] without knowledge of the structural damping and contact damping and calculates the fitness function exclusively based on the impedance modulus curve. In Method 1, the six unknown parameters to be identified in the impedance equations were set as: $\partial =[\;\mu_{33}^{\prime }\;\mu_{33}^{\prime \prime }\;S_{33}^{\prime }\;S_{33}^{\prime \prime }\;d_{33}^{\prime }\;d_{33}^{\prime \prime }{]}^{T}$. Equation (11) is used to generate the simulated electric impedances and it is expressed as $\hat{Z}(i,\partial )$. The experimentally measured impedance modulus values are expressed as Z(i). The RMSE of the difference between $\hat{Z}(i,\partial )$ and the Z(i) is taken as the objective function $E_{1}(\partial )$, which is shown in Equation (14). The six values identified by the PSO are used as the final result of material parameter extraction of the Method 1.
(14)$$F_{z}(i)=E_{1}(\partial )=\sqrt{\frac{1}{I}\sum_{i}^{I}{\left(Z(i)-\hat{Z}(i,\partial )\right)}^{2}}$$

In Method 2, the structural damping and contact damping are still unknown. Different from Method 1, the fitness function value of Method 2 is calculated based on the phase angle data, and the imaginary parts of the complex parameters are extracted by PSO instead. The real parts of the complex parameters are still determined by Method 1. The three unknown parameters to be identified in the impedance equations are set as $\delta =[\;\mu_{33}^{\prime \prime }\;S_{33}^{\prime \prime }\;d_{33}^{\prime \prime }{]}^{T}$. Equation (12) is used to generate the simulated phase and it is expressed as $\hat{P}(i,\delta )$. The experiment measured phase are expressed as P(i). The RMSE of the difference between $\hat{P}(i,\delta )$ and the P(i) is taken as the objective function $E_{2}(\delta )$, which is shown in Equation (15).
(15)$$F_{p}(i)=E_{2}(\delta )=\sqrt{\frac{1}{I}\sum_{i}^{I}{\left(P(i)-\hat{P}(i,\partial )\right)}^{2}}$$

In Equations (14) and (15), *I* represents the total number of sampling points and *i* represents the *i*-th sampling point.

In Method 3, the structural damping and contact damping of the transducer are measured, calculated, and entered as known values into the parameter identification process. The settings for both the unknown parameters and fitness function, as well as the algorithm iteration process, are the same as in Method 2.

## 3. Experimental Measurement

### 3.1. The Measurement of the Displacement Plunger’s Structural Damping

The damping ratios of stainless steel 304 bars were measured through standard modal testing and then calculated by the structural damping of the displacement plunger [32,33]. A standard cylindrical sample with a diameter of 50 mm and a length of 1500 mm was prepared. At this size, the natural frequency for the first longitudinal vibration of the sample can be close to the working frequency of the transducer. The test equipment is shown in Figure 6. The piezoelectric accelerometers, as well as the real-time data measurement and analysis instrument DH5922D and its analysis software, were sourced from Donghua Testing Technology Co., Ltd (Taizhou, China). The sample bar was hung with a soft elastic rope and the suspension point was located on the modal node of the sample. A levelling instrument was used to check whether the rod was level. A force hammer was used to apply a force on one end of the bar, and four piezoelectric accelerometers were installed on the other end to measure the resulting acceleration.

The schematic block diagrams of the standard modal experiment are shown in Figure 7. A 5 kN test hammer was used to act on the tested 304 stainless steel bar to vibrate the sample, and the vibration signal was picked up by the piezoelectric accelerometer and converted into an electrical signal. The charge amplifier amplifies the electrical signal, filters it, and then performs analog/digital (A/D) signal conversion, which is implemented in the instrument DH5922D. The time domain signal is subjected to fast Fourier transform (FFT), and then the frequency response function (FRF) is calculated, and, finally, the damping ratio *ξ_d_* is obtained by parameter identification.

Figure 8 shows the real and imaginary parts of the FRF of the sample. It can be seen that the first longitudinal resonance frequency *ω*_1_ of the sample is 1600 Hz. Using the widely used poly-reference least squares complex frequency domain method (PolyMax) for modal parameter identification, the damping ratio *ξ_d_* at the natural frequency of the first-order longitudinal vibration of the material was extracted to be 0.294%, and, according to the JCHM standard [34,35], the standard uncertainty is 0.019%. It is known that the mass of the displacement plunger (shown in Figure 3) is *M_d_* = 0.722 kg, so the structural damping of the displacement plunger is *R_d_* = 2*ξ_d_M_d_ω*_1_ = 6.79 N/(m/s) and the standard uncertainty is 0.4390 N/(m/s).

### 3.2. The Contact Damping Calculation

In transducers, contact damping has a great impact on the performance [32,36]. According to [32], these losses account for 45% of the total mechanical losses in a typical piezoelectric transducer. Consequently, it is necessary to calculate the contact damping of the contact surfaces within the transducer. There are three main contact surfaces, namely, the contact surface of the giant magnetostrictive rod and the magnetic column (G-M surface), the magnetic column and the displacement plunger (M-D surface), and the magnetic column and the housing base (M-H surface).

#### 3.2.1. Morphology of the Rough Surfaces

The surfaces of the giant magnetostrictive rod (GMR), magnetic column (iron), and displacement plunger (304 stainless steel) were polished with 240-mesh, 600-mesh, and 800-mesh sandpaper in order. Then, a Bruker’s Contour Elite 3D microscope was used to measure the rough surface morphology (Figure 9). In order to ensure the accuracy of the results, six different points (size of each point of 2538.46 μm × 1903.84 μm) were selected for the surface topography characterization and the roughness parameters of the six positions were averaged.

The measurement data and the standard uncertainty of measurement are displayed in Table 1. For a three-dimensional rough surface, *R*_q_ represent the RMSE of roughness.

According to [37,38], the fractal dimension *D* and the fractal roughness parameter *G* of the rough surface profile can be calculated by Formulae (16)–(18),
(16)$$D_{s}=\frac{1.548}{R_{q}^{0.041}}$$
(17)$$D=D_{s}+1$$
(18)$$G=\sqrt[{2(D_{s}-1)}]{R_{q}^{2}\sin[\frac{\pi (2D_{s}-3)}{2}]\Gamma (2D_{s}-3)/L_{m}^{\left(2-D_{s}\right)}}$$
where Γ(x) is Euler’s gamma function, *L*_m_ is the length of the specimen for the surface profile. The calculation results are shown in Table 2.

#### 3.2.2. Calculation of the Contact Damping

According to Hertz contact theory, the contact between two rough surfaces can be modeled as a contact between a rigid plane and an equivalent rough surface [39]. Mandelbrot et al. used the W-M function to apply fractal geometry theory to the contact analysis between two rough surfaces, and established a new fractal contact model (M-B model). So far, many scholars have conducted extensive research on the contact problem of rough surfaces based on the fractal model [40,41,42].

In this study, the contact damping in the transducer was calculated based on the fractal model. It is known that the fractal dimension and the fractal roughness parameter of two rough surfaces are *D*_x_ and *G*_x_, where x = 1 and 2, respectively. When two rough surfaces contact, the equivalent fractal dimension *D*_eq_ and the equivalent fractal roughness parameter *G*_eq_ of the contact surfaces are defined as follows: *D*_eq_ = max (*D*_1_, *D*_2_) and $G_{eq}=\sqrt[{2(D_{eq}-1)}]{G_{1}^{2(D_{1}-1)}+G_{2}^{2(D_{2}-1)}}$.

The detailed calculation process is presented in Appendix A and the results and standard uncertainty for a pre-stress of 10 Mpa are shown in Table 3. As the contact damping calculation equation is nonlinear, it is more appropriate to use a Monte Carlo method (MCM) of JCGM [34] to evaluate the uncertainty. The evaluation results are shown in Table 3, where the size of the Monte Carlo test sample is M_1_ = 10^6^.

### 3.3. The Measurement of Impedance/Phase

The impedance/phase measurement device is shown in Figure 10. The base of the longitudinal vibration transducer was fixed on a shock absorbing table with screws to simulate the vibration state of being clamped at one end. An impedance analyzer was used to measure the impedance and phase data of the longitudinal transducer and a ring pressure sensor was used to measure the value of the pre-stress applied to the magnetostrictive rod. A programmable DC power supply was used to provide 2 A DC current for the purpose of magnetic bias.

### 3.4. Estimated Standard Uncertainty and Expanded Uncertainty of Measurement

In summary, both the estimated standard uncertainty and expanded uncertainty U99 of the measurements are shown in Table 4, where P represents the confidence probability and K represents the confidence factor. The uncertainty closes to a uniform distribution, where when P is 99%, the corresponding K is 2.58.

## 4. Experimental Results

### 4.1. Sensitivities Analysis

The quality of parameter identification was first evaluated by global sensitivity analysis. The goal of global sensitivity analysis is to determine the importance of parameters with interaction effects between multiple parameter variances. This means that if the fitness function value is more sensitive to one parameter, this identified parameter is closer to its true value. Usually, without knowing the actual parameters, sensitivity analysis allows rational choices under uncertain conditions [43,44].

Scatterplots (Figure 11 and Figure 12) were used to analyse the global sensitivity while considering the interactions between the variance of the six parameters. The abscissa represents the parameter variances of each iteration by PSO and the ordinate represents the fitness function values obtained in each iteration. The parameter values were divided by the average number for normalization. The red inverted triangles are used to indicate the local minimum. The sharpness of the basin in the scatterplots reflects the sensitivity of the fitness function to the complex parameters. The sharpness can be quantified by ΔF, where ΔF is the width of the basin when the fitness function value is equal to 1.5. With a smaller ΔF, a higher sensitivity of the fitness function to a specific parameter is demonstrated [14].

#### 4.1.1. Sensitivities of the Method 1

The global sensitivity of each parameter in Method 1 is shown in Figure 11. It is apparent from Figure 11a–c that the fitness function is highly sensitive to $\mu_{33}^{\prime }$, $S_{33}^{\prime }$, and $d_{33}^{\prime }$; however, the fitness function is far less sensitive to $\mu_{33}^{\prime \prime }$, $S_{33}^{\prime \prime }$, and $d_{33}^{\prime \prime }$, (Figure 11d–f). The basins of the scatterplots are almost planar, and the ΔF_1_ values corresponding to each of the three imaginary parts are about 10 times those of the corresponding real part, indicating that the losses extracted by Method 1 are unreliable.

#### 4.1.2. Sensitivities of the Method 2 and 3

The sensitivities of each parameter in Methods 2 and 3 are shown in Figure 12.

For Method 2 (the gray spots), the fitness function value was highly sensitive to $\mu_{33}^{\prime \prime }$, but the sensitivities to $S_{33}^{\prime \prime }$ and $d_{33}^{\prime \prime }$ were very low, indicating that these two losses cannot be identified by Method 2. In Method 3 (the blue spots), the sensitivities of $S_{33}^{\prime \prime }$ and $d_{33}^{\prime \prime }$ were greatly improved (Figure 12b,c) and the ΔF_3_ values of each parameter were about ten times smaller than ΔF_2_. This is because after calculating the structural damping and contact damping of the transducer, the searching range used by the PSO algorithm in Method 3 is greatly reduced when compared with the search range in Method 2, so the fitness function is greatly improved for the sensitivity to elastic losses and piezomagnetic coupling losses. In summary, Method 3 can extract more reliable material losses.

#### 4.1.3. Uncertainty of Damping and Sensitivity Analysis

The influence of the uncertainty of structural damping and contact damping on the sensitivity of parameter identification was evaluated. It can be seen from Table 4 that the structural damping of the displacement plunger features an interval of 6.79 ± 1.1326 N/(m/s) with a probability of 99%, and the contact damping of the rough surface features an interval of 324.5469 ± 7.8081 N/(m/s) with a probability of 99%. Consequently, it is necessary to evaluate the influence of the uncertainty of structural damping and contact damping on the sensitivity of parameter identification. In this study, it is proposed to input the upper and lower limits of the confidence interval of damping into Method 3 in order to analyze the parameter sensitivity. The statistics for ΔF are shown in Table 5.

From Table 5, the uncertainty of damping has almost no effect on the sensitivity of the imaginary parts of the material complex parameters.

### 4.2. Comparison between Simulation and Experiments

Figure 13a is the comparison of the experimental impedance modulus data and the simulation data of the transducer (Figure 1) at 10 Mpa, and Figure 13b is the comparison of the experimental phase data and the simulation data. Table 6 shows the RMSE and determination coefficient (*R*^2^) between the experimental data and simulation data.

Method 1 is based on the impedance modulus data for parameter extraction. When using the parameters extracted by Method 1, the obtained simulated impedance modulus is highly consistent with the experimental curve (Figure 13a). The value of the RMSE is 1.8589 and *R*^2^ is 0.9963, which indicates that the similarity between the simulation data and the experimental data is 99.63%; however, the phase angle curve simulation is significantly deviated from the measured data, and the RMSE and *R*^2^ values are 6.2623 and 0.4573, respectively. This shows that if Method 1 is used, even if the fitness function reaches a minimum via PSO, the extracted losses might be abnormal.

Method 2 is based on the phase data for parameter extraction. When using the parameters extracted by Method 2 for simulation, the simulated phase curve is consistent with the experimental curve (Figure 13b), and the value of the RMSE is 2.5243 and *R*^2^ reaches 0.9170; however, it can be seen from Figure 13a that the simulated impedance minimum point and impedance maximum point data have great differences from the experimental data, where the RMSE is 3.2979, which is about two times that of Method 1 (1.8589) and Method 3 (1.3377). The losses of the material cannot be reliably characterized with this approach.

When the parameters extracted by Method 3 were used for simulation, the simulated results of both impedance modulus and phase were very consistent with the experimental data, and the *R*^2^ values were 0.9981 and 0.9746, respectively. This successfully verified the effectiveness of Method 3.

### 4.3. Stability and Uncertainty

A PSO algorithm is a stochastic population-based optimization algorithm, and the initial values of parameters are randomly selected within the search range. As such, stability is also one of the important performances to measure the quality of parameter identification [45]. For Methods 1–3, ten random parameter extractions were performed and the results were compared. The coefficient of variation was used to quantify the relative uniformity, which is defined as cv=(σ/μ)×100%. *σ* represents the standard deviation and *μ* represents the average of this set of data. The smaller the coefficient of variation, the better the stability of the parameter identification results. The results of the ten-time parameter extraction by Methods 1–3 are shown in Figure 14.

The settings of the PSO algorithm were kept consistent in the three methods. The size of the particle swarm was 12, the learning factor was *c*_1_ = *c*_2_ = 2, and the inertia weight ω was 0.5. Table 7 summarizes the search range of the six parameters, the means ($\bar{x}$), and absolute values of the coefficient of variation (|cv|) for the ten-time identification results.

The results show that the repeatability of the real parts extracted by Methods 1–3 is satisfactory. In Methods 1 and 2, the coefficients of variation for the imaginary parts are large, which indicates very poor repeatability. This is because Methods 1 and 2 do not consider the structural damping and contact damping of the transducer, which leads to a large search range for the three parameters of $\mu_{33}^{\prime \prime }$, $S_{33}^{\prime \prime }$, and $d_{33}^{\prime \prime }$. The fitness function has low sensitivity to them. As such, the identification results of PSO algorithm have strong randomness and low repeatability.

As shown in Figure 14, the three losses extracted by Method 3 are very stable. Figure 14a shows the poor stability of the hysteresis losses extracted by Methods 1 and 2. Figure 14b,c shows that the elastic losses and piezomagnetic coupling losses extracted by Methods 1 and 2 switch between positive and negative values, which was similarly observed in [21,22,23]; however, there is no case where the imaginary parts turn positive under small signal excitation [24]. It is worth noting that the losses extracted by Method 3 stay negative and the coefficient of variation is very low, indicating that the material losses stably reached true values.

The uncertainty is used to characterize the dispersion of the results. The smaller the relative uncertainty, the higher the credibility of the parameter extraction results. According to the MCM and the uncertainty synthesis method in [35], the relative uncertainty of the parameters extracted by Method 3 was evaluated and the results are shown in Table 8. The parameters output by the MCM are close to a normal distribution, where, when P is 99%, the corresponding K is 3.

## 5. Conclusions

In this paper, a novel optimization method to characterize the complex parameters of GMMs under pre-stress has been presented. By measuring and calculating the structural damping and contact damping, the ranges of parameter variance can be well confined. Then, the real parts of the three complex parameters are characterized by the impedance modulus data while the imaginary parts are characterized by the phase data. The PSO algorithm is adopted to minimize the fitness function values.

The global sensitivity analysis shows that the proposed method remarkably improves the sensitivity of three losses to the fitness function compared with the conventional method, which strongly implies the success of true value extraction. The uncertainty values of the six complex parameters $\mu_{33}^{\prime },\;S_{33}^{\prime },\;d_{33}^{\prime },\;\mu_{33}^{\prime \prime },\;S_{33}^{\prime \prime }$ and $d_{33}^{\prime \prime }$ are 0.138%, 0.048%, 0.330%, 4.215%, 7.026%, and 12.037%, respectively. The coefficients of determination for *R*_2_ between the experimental data and simulation data were 99.81% (impedance data) and 97.46% (phase data).

Additionally, the ten-time random parameter characterization results show that the proposed method has high repeatability and produces parameters that feature greater reliability. The comparison between the simulated impedance/phase data and the experimental shows that the simulation of the proposed lump parameter circuit model using the parameters extracted by the proposed method can simultaneously reproduce the impedance and phase data with high fidelity. In the future, more experimental measurements and complex parameter characterizations under different working conditions can be carried out with our proposed method, which can provide key data support for the research and design of giant magnetostrictive transducers.

## Figures and Tables

**Figure 1 micromachines-12-01416-f001:**
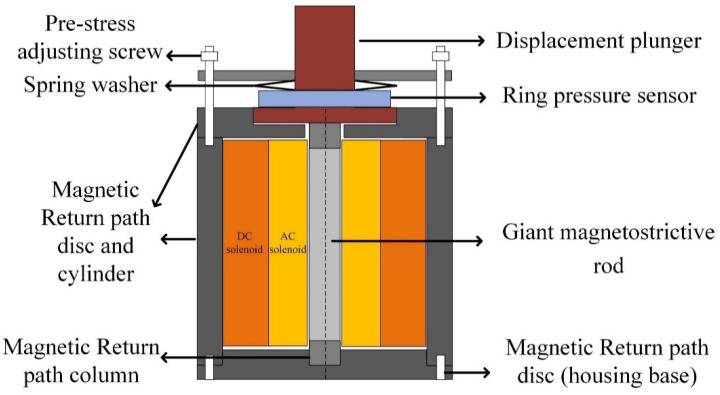
Structural diagram of the longitudinal transducer.

**Figure 2 micromachines-12-01416-f002:**
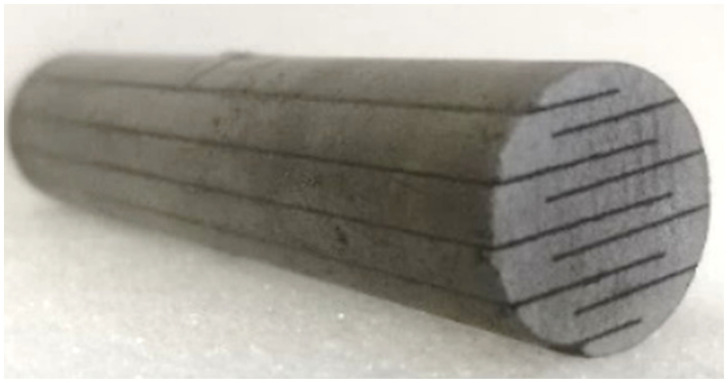
Photo of GMM rod for experiment.

**Figure 3 micromachines-12-01416-f003:**
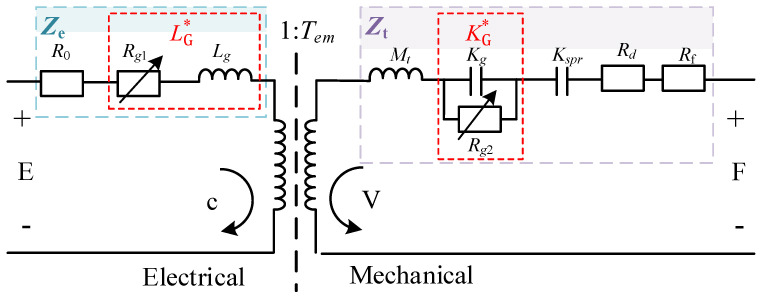
Schematic illustration of improved lumped parameter model of the transducer.

**Figure 4 micromachines-12-01416-f004:**
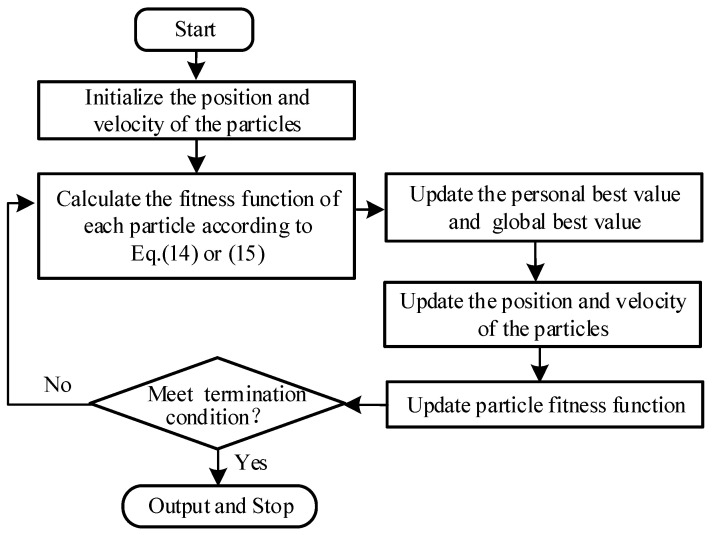
Flowchart of the PSO.

**Figure 5 micromachines-12-01416-f005:**
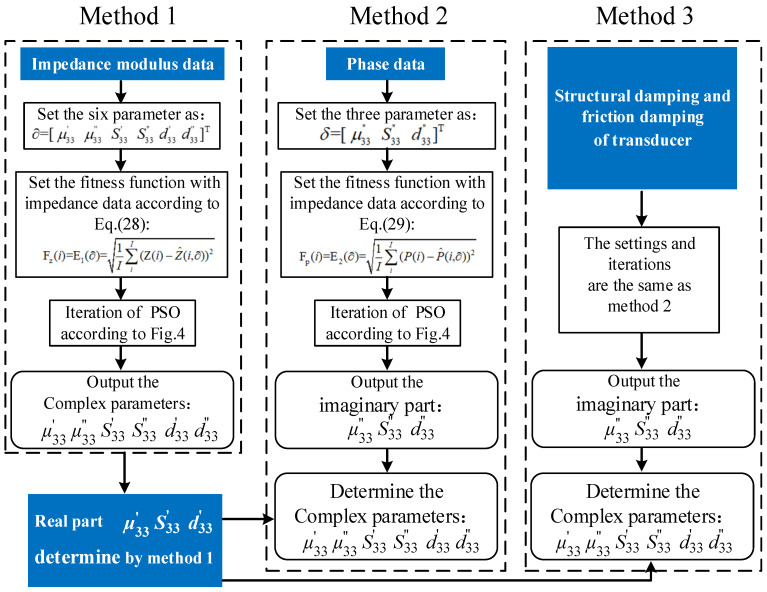
Flowchart of three methods of parameters characterization.

**Figure 6 micromachines-12-01416-f006:**
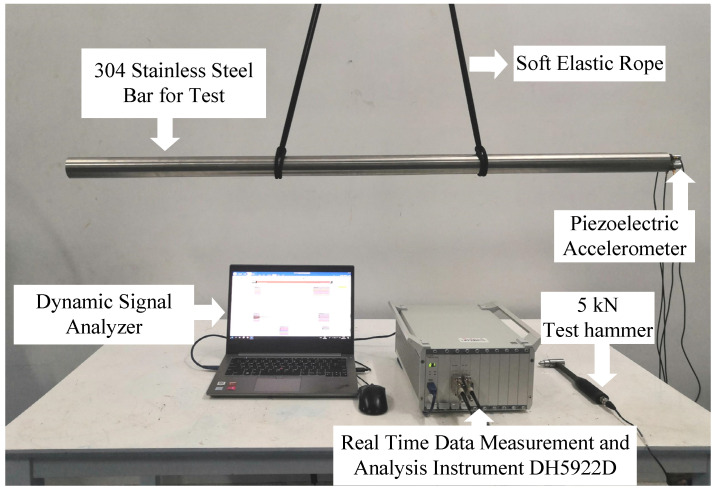
Experimental device for extracting the damping ratio of a stainless steel 304 bar sample.

**Figure 7 micromachines-12-01416-f007:**
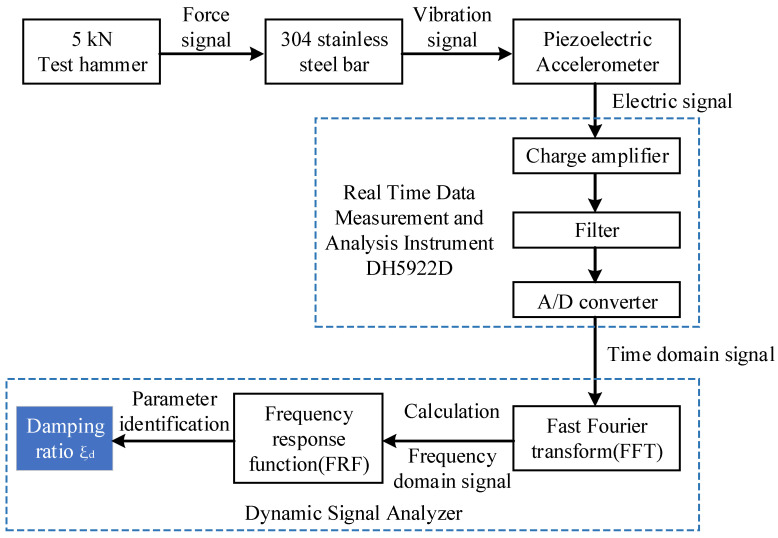
Schematic block diagrams of standard modal test.

**Figure 8 micromachines-12-01416-f008:**
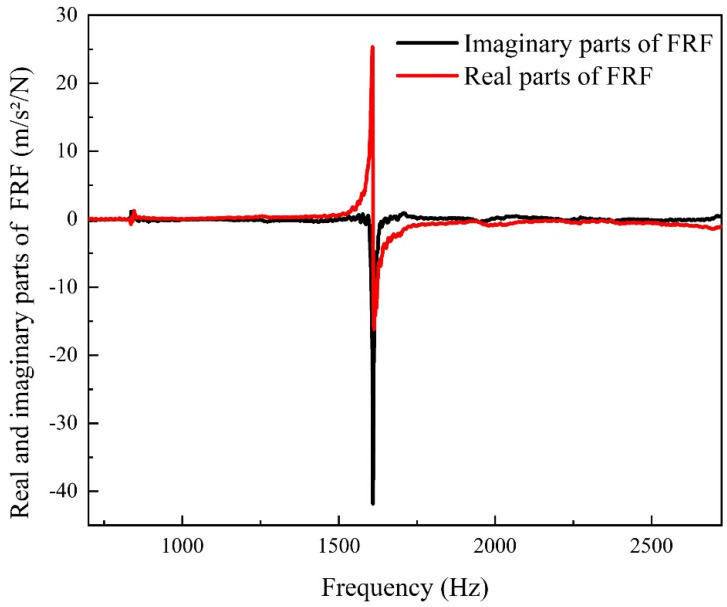
FRF of the stainless steel 304 bar sample.

**Figure 9 micromachines-12-01416-f009:**
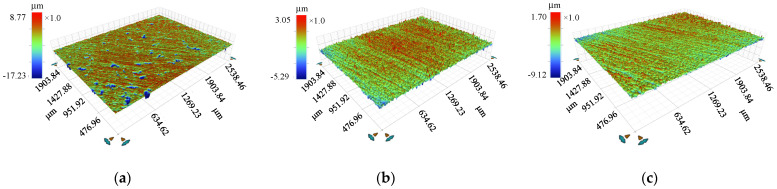
Rough surface morphology for the (**a**) GMR, (**b**) magnetic column, (**c**) and displacement plunger.

**Figure 10 micromachines-12-01416-f010:**
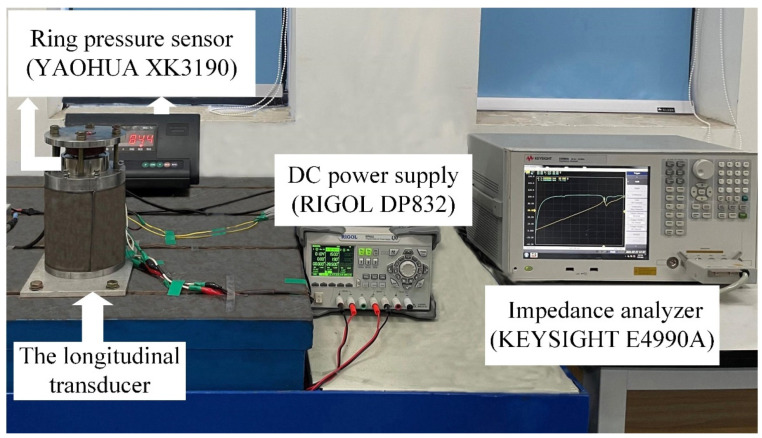
The impedance/phase measurement device.

**Figure 11 micromachines-12-01416-f011:**
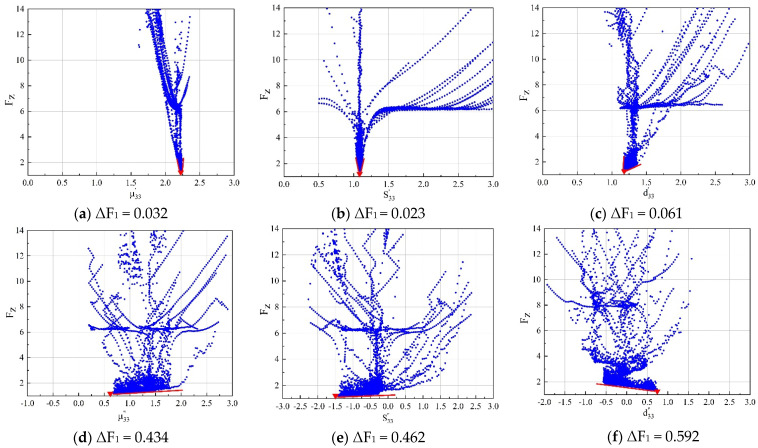
Sensitivities of six parameters in Method 1. ΔF_1_ is used to quantify the global sensitivity, which has been defined.

**Figure 12 micromachines-12-01416-f012:**
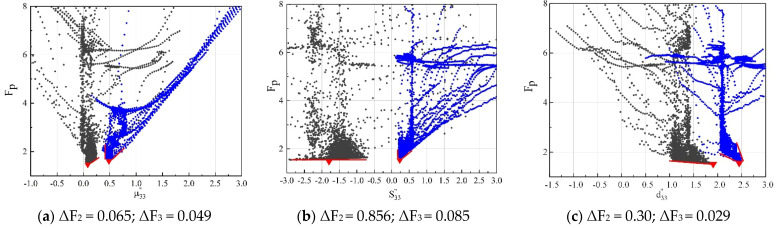
Sensitivities of three imaginary parts in Method 2 (the gray spots) and Method 3 (the blue spots). ΔF_2_ and ΔF_3_ are used to quantify the sensitivity of each parameter in Methods 2 and 3, respectively.

**Figure 13 micromachines-12-01416-f013:**
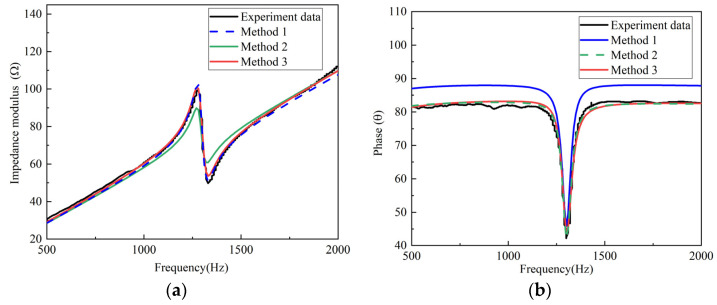
Comparison of the experimental data and simulation data under pre-stress conditions of 10 Mpa (**a**) impedance modulus and (**b**) phase.

**Figure 14 micromachines-12-01416-f014:**
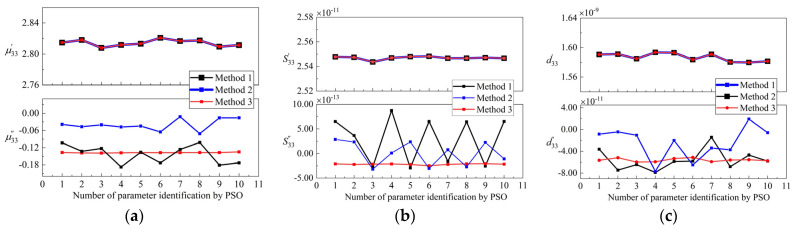
Comparison of the results of ten extractions with the three different methods at 10 Mpa (**a**) $\mu_{33}^{\prime }\;\mathrm{and}\;\mu_{33}^{\prime \prime }$; (**b**) $S_{33}^{\prime }{\;\mathrm{and}\;S}_{33}^{\prime \prime }$; (**c**) $d_{33}^{\prime }\;\mathrm{and}\;d_{33}^{\prime \prime }$.

**Table 1 micromachines-12-01416-t001:** Measurement parameters *R*_q_ and standard uncertainty of rough surfaces.

	1	2	3	4	5	6	Ave.	Standard Uncertainty
*R*_q_ (μm) of the GMR	0.577	0.490	0.539	0.706	0.910	0.744	0.661	0.012
*R*_q_ (μm) of the magnetic column	0.345	0.307	0.333	0.394	0.368	0.338	0.347	0.011
*R*_q_ (μm) of the displacement plunger	0.315	0.378	0.338	0.308	0.303	0.297	0.323	0.011

**Table 2 micromachines-12-01416-t002:** Measurement parameters of the rough surfaces.

	Rough Surfaces of the GMR	Rough Surfaces of the Magnetic Column and Housing Base	Rough Surfaces of the Displacement Plunger
*D*	2.5747	2.6167	2.6214
*G* (m)	2.2525 × 10^−10^	2.8175 × 10^−10^	2.8890 × 10^−10^

**Table 3 micromachines-12-01416-t003:** Parameters (estimated value ± standard uncertainty) of contact surfaces with a pre-stress of 10 Mpa.

	Contact Damping (N/(m/s))	*D* _eq_	*G*_eq_ (m)
G-M surface	36.5693 ± 0.1558	2.6167 ± 0.0011	4.3788 × 10^−10^ ± 9.6892 × 10^−12^
M-D surface	287.9776 ± 2.8382	2.6214 ± 0.0009	3.6296 × 10^−10^ ± 2.8896 × 10^−13^
M-H surface	24.6834 ± 0.0324	2.6167 ± 0.0011	3.4991 × 10^−10^ ± 2.4283 × 10^−13^
Sum	349.2303 ± 3.0274	/	/

**Table 4 micromachines-12-01416-t004:** Estimated uncertainty of measurements.

	Measurement of Structural Damping	Measurement of Contacting Damping	Measurement of Impedance and Phase
Standard Uncertainty (P = 68.27%, K = 1)	0.4390 N/(m/s)	3.0264 N/(m/s)	0.00046 Ω/0.0058 deg
Expanded Uncertainty (P = 99%, K = 2.58)	1.1326 N/(m/s)	7.8081 N/(m/s)	0.00118 Ω/0.0149 deg

**Table 5 micromachines-12-01416-t005:** The value of ΔF when the damping is in the confidence interval (P = 99%).

	$\mu_{33}^{\prime \prime }$	$S_{33}^{\prime \prime }$	$d_{33}^{\prime \prime }$
Damping is the mean of the confidence interval	0.049	0.085	0.029
Damping is the upper limit of the confidence interval	0.051	0.084	0.031
Damping is the lower limit of the confidence interval	0.048	0.082	0.027

**Table 6 micromachines-12-01416-t006:** The RMSE and *R*^2^ between the experimental data and simulation data.

	Method 1	Method 2	Method 3
RMSE of impedance modulus data	1.8589	3.2979	1.3377
RMSE of phase data	6.2623	2.5243	1.6748
*R*^2^ of impedance modulus data	0.9963	0.9887	0.9981
*R*^2^ of phase data	0.4573	0.9170	0.9746

**Table 7 micromachines-12-01416-t007:** Extraction results and the coefficients of variation for six parameters.

Parameter	Search Range	Method 1		Method 2		Method 3	
		$\bar{x}$	|cv|%	$\bar{x}$	|cv|%	$\bar{x}$	|cv|%
$\mu_{33}^{\prime }$	[2, 3]	2.814	1.460 × 10^−3^	\	\	\	\
$S_{33}^{\prime }$	[1 × 10^−12^, 1 × 10^−10^]	2.547 × 10^−11^	5.013 × 10^−4^	\	\	\	\
$d_{33}^{\prime }$	[5 × 10^−10^, 5 × 10^−9^]	1.587 × 10^−9^	3.420 × 10^−3^	\	\	\	\
$\mu_{33}^{\prime \prime }$	[−0.3, 0.3]	−0.143	0.224	−0.040	0.507	−0.137	0.008
$S_{33}^{\prime \prime }$	[−1 × 10^−12^, 1 × 10^−12^]	−2.861 × 10^−13^	1.653	−7.170 × 10^−15^	33.818	−2.178 × 10^−13^	0.052
$d_{33}^{\prime \prime }$	[−9 × 10^−11^, 9 × 10^−11^]	−5.581 × 10^−11^	0.343	−2.435 × 10^−11^	1.205	−5.588 × 10^−11^	0.055

**Table 8 micromachines-12-01416-t008:** Relative uncertainties of material complex parameters extracted by Method 3.

	$\mu_{33}^{\prime }$	$S_{33}^{\prime }$	$d_{33}^{\prime }$	$\mu_{33}^{\prime \prime }$	$S_{33}^{\prime \prime }$	$d_{33}^{\prime }$
Relative standard uncertainty (P = 68.27%, K = 1)	0.046%	0.016%	0.110%	1.405%	2.342%	4.009%
Relative expanded uncertainty (P = 99%, K = 3)	0.138%	0.048%	0.330%	4.215%	7.026%	12.037%

## Data Availability

Data are contained within the article.

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
