# Peer review of "Characterization of Giant Magnetostrictive Materials Using Three Complex Material Parameters by Particle Swarm Optimization"

_micromachines, 2021, doi:10.3390/mi12111416_

Round 1

Reviewer 1 Report

In the Review

Author Response

Dear Prof,

Thank you very much for your affirmation and support of our manuscript.

We will continue to improve future research work.

With best wishes,

Prof. Xin Yang

Reviewer 2 Report

The paper can be accepted after the following corrections:

  1. Please provide schematic block diagrams of measuring setups presented in figures 6 and 7.
  2. Uncertainty of measurements should be estimated.
  3. The coefficient of variation of results of optimization was presented. However, please present also the statistical results of analyze of accuracy of comparison between experimental data and the results of modelling.
  4. Please develop the conclusions to clearly present the most important quantitative data as well as please present the outlook for further research.

Round 2

Reviewer 2 Report

The paper was corrected and can be accepted in the present form.